# Scalable Structure for Chiral Quantum Routing

**DOI:** 10.3390/e27050498

**Published:** 2025-05-05

**Authors:** Giovanni Ragazzi, Simone Cavazzoni, Claudia Benedetti, Paolo Bordone, Matteo G. A. Paris

**Affiliations:** 1Dipartimento di Scienze Fisiche, Informatiche e Matematiche, Università di Modena e Reggio Emilia, I-41125 Modena, Italy; simone.cavazzoni@unimore.it (S.C.); paolo.bordone@unimore.it (P.B.); 2Dipartimento di Fisica Aldo Pontremoli, Università di Milano, I-20133 Milano, Italy; claudia.benedetti@unimi.it (C.B.); matteo.paris@fisica.unimi.it (M.G.A.P.); 3Centro S3, CNR-Istituto di Nanoscienze, I-41125 Modena, Italy

**Keywords:** quantum walks, routing on graphs, chirality

## Abstract

We address the problem of routing quantum and classical information from one sender to many possible receivers in a network. By employing the formalism of quantum walks, we describe the dynamics on a discrete structure based on a complete graph, where the sites naturally provide a basis for encoding the quantum state to be transmitted. Upon tuning a single phase or weight in the Hamiltonian, we achieve near-unitary routing fidelity, enabling the selective delivery of information to designated receivers for both classical and quantum data. The structure is inherently scalable, accommodating an arbitrary number of receivers. The routing time is largely independent of the network’s dimension and input state, and the routing performance is robust under static and dynamic noise, at least for a short time.

## 1. Introduction

Quantum communication and quantum computation rely on the accurate transmission and routing of quantum information [1,2,3,4,5,6,7]. A robust and efficient transfer of a quantum state between two distant but specified components of a larger quantum system (i.e., a quantum computer or a quantum channel) is a fundamental requirement for practical quantum computation and communication [8,9]. Developing appropriate and scalable protocols for quantum routing remains a crucial and cutting-edge research area. A testbed platform for quantum routing can be implemented using a continuous-time quantum walk (QWs) evolving on a network [10]. Indeed, quantum walks naturally describe the evolution of a quantum particle over a set of discrete positions, i.e., the nodes of a graph, pairwise connected by edges. QWs have emerged as a fundamental framework for numerous applications, including quantum transport [11,12,13,14,15], quantum computation [16,17,18], quantum algorithms [19,20,21,22,23], and network estimation and characterization [24,25,26,27]. In recent years, the impact of chirality [28,29,30,31] on quantum walk dynamics has been investigated, and exploited to improve quantum information tasks. Chirality adds new degrees of freedom into the Hamiltonian, that can be employed to obtain a quantum advantage [32,33,34,35,36,37]. Indeed, a chiral Hamiltonian has complex off-diagonal elements, which introduce directionality to the system, while still fulfilling the Hermitian condition essential for the system’s unitary evolution. The implementation of these systems in experimental settings has been explored in recent research, particularly within the framework of synthetic gauge fields in lattice structures [38,39,40].

In this work, we propose a quantum routing scheme that exploits the chirality of a continuous-time quantum walk to enhance the fidelity of quantum state transfer. The core of the routing architecture is modeled as a complete graph connecting an input node to multiple output ports. Our aim is to achieve on-demand quantum information transfer to a specific output port, with information encoded either in a single node or as a superposition of vertices. By exploiting the intrinsic symmetries of the network, we show that the proposed quantum router maintains high efficiency even in the presence of a very large number of output ports.

The realistic implementations of quantum walks must account for the effects of noise and defects, which can lead to decoherence, localization, and the loss of quantum properties [41,42,43,44,45,46]. Defects can be modeled as random variations in certain Hamiltonian parameters, while classical noise manifests as stochastic fluctuations. We thus investigate the impact of defects and noise on the chiral phase of the quantum walk. Specifically, defects are modeled as random variables drawn from a von Mises distribution [47], while noise arises from an Ornstein–Uhlenbeck process [48,49]. Our analysis reveals that both phenomena affect the dynamics of the fidelity of the transferred state. However, high fidelity is preserved at short times.

The paper is organized as follows. In Section 2, we review the theoretical framework underlying quantum routing within the quantum walk formalism. In Section 3, we evaluate the performance of the quantum router for classical information, comparing two distinct strategies: the chiral protocol and a protocol based on link weighting. In Section 4, we examine the transmission of quantum information across three different configurations with varying numbers of output ports, demonstrating that fidelity remains close to unit. Subsequently, in Section 5, we investigate the influence of defects and noise on the chiral phase and their effects on the routing process. Section 6 closes the paper with some concluding remarks.

## 2. Quantum Routing with Quantum Walks

Quantum walks describe the evolution of a quantum particle, referred to as the walker, over a discrete position space, mathematically represented as a graph. In particular, we consider the evolution of a QW over a connected simple graph G(V,E), where V={1,…,N} is the set of the nodes (or vertices) and E specifies the links (or edges) between the nodes of the graph. Let us call *N* the cardinality N=|V| of such a graph. In order to describe the quantum evolution of the walker over G, the natural choice is to define the orthonormal site basis {|j〉}j=1N, where |j〉 denotes a quantum walker localized at the jth vertex. Assuming the system to be isolated and setting ℏ=1, the dynamics is governed by the time evolution operator U(t)=e−iHt, where *H* is an Hamiltonian that describes the structure of the graph G and *t* is an effective time absorbing both *ℏ* and any scaling constant that could appear in front of *H*. A common choice for the QW Hamiltonian is the adjacency matrix of the graph which encodes the topology of the graph and its elements are(1)Ajk=1ifj≠kand(j,k)∈E0otherwise,However, when dealing with chiral quantum walks, the Hamiltonian governing the system must incorporate complex terms to account for direction-dependent phase factors. These complex contributions break time-reversal symmetry and play a crucial role in determining the walker’s propagation dynamics [30,31]. It follows that the elements of the chiral Hamiltonian takes the general form Hjk=Ajkeiϕjk, where ϕjk represents the chiral phase associated with the directed edge (j,k). The requirement of Hermiticity imposes the constraint ϕkj=−ϕjk, ensuring that the Hamiltonian remains physically valid. This phase asymmetry is a defining feature of chiral quantum walks, leading to nontrivial transport properties over the output nodes and directional bias in the quantum walker evolution.

The problem of quantum routing consists of directing quantum information from a designated set of input nodes to one of multiple possible output ports, with the specific target selected through an appropriate control mechanism. A fundamental requirement of any quantum routing protocol is the preservation of coherence and high-fidelity transmission. In addition to fidelity, several key factors must be considered, including the computational and physical cost associated with switching between the target nodes, the time required for the routing process to complete, and the duration for which the overlap between the evolved input state and target one remains sufficiently high. The effectiveness of chiral QW for routing purposes was already investigated in [35], for a router with one input and two possible outputs. Tuning the phase distributed along the links of the internal loop, it was demonstrated that it is possible to obtain a high probability of finding the walker in one of the two outputs. In the present work, we generalize these results designing a router with *n* outputs.

We propose a quantum routing scheme based on the graph depicted in Figure 1a. The core of the network consists of a complete (n+1)-node graph, where each node is additionally connected via a single edge to a peripheral vertex that serves either as input or output node. This structure enables efficient state transfer by harnessing the connectivity of the complete graph.

If the task involves routing classical information or a quantum spin state in a single excitation subspace [50,51], it is enough to consider transition probabilities between localized states. In this case, one external vertex acts as input and the other *n* serve the output ports (due to the symmetry of the network, any input or output can assume the role of the sender or the receiver depending on the circumstances). To route quantum information by exploiting quantum coherence among a set of input nodes, the input and output states are instead the quantum superpositions of an external vertex and its corresponding connected internal vertex, as can be seen in Figure 1a.

If the dynamics of the routing structure is governed by the adjacency matrix of the graph, i.e., H=A, the Hamiltonian is real and symmetric, which prevents the routing of information due to the lack of directional bias. To enhance state transfer across the network, we introduce a chiral phase ϕ∈[0,2π) and an edge weight β∈R on the single link connecting the internal nodes of the complete graph associated to the input and output vertices (the star labeled edge between the cyan and purple sites in Figure 1a). For example, if we denote by |x¯j〉 the internal node corresponding to the initial state, whether localized or in a superposition, and |x¯k〉, the internal node corresponding to an output state, the corresponding Hamiltonian element is given by 〈x¯j|H|x¯k〉=βeiϕ. To maintain Hermiticity, we need 〈x¯k|H|x¯j〉=βeiϕ. Explicitly, naming the vertices of the complete graph |x¯m〉 and the external vertices of the structure |y¯m〉, the associated Hamiltonian matrix of the structure is given by(2)H=∑mn+1∑l<mn+1|xm〉〈xl|+∑mn+1|xm〉〈ym|−1−βeiϕ|x¯j〉〈x¯k|+h.c.
where the first sum is associated to the adjacency of the internal complete graph, and the second sum to the connections between the core of the structure and the external vertices. The final term, instead, breaks the symmetry and allows the routing procedure in the j↔k direction. One advantage of this model is that, when optimizing the phase or the weight to route a state from the input to a specific output port, the structure of the routing network ensures that changing that the sender or the receiver requires only applying the same phase or weight to a different link, corresponding to the new set of nodes. This simplifies the process of reconfiguring the network for different routing scenarios. In addition to the ability to select both the sender and receiver, the structure of our network can further be exploited to perform dimensionality reduction [52,53]. This allows us to map the routing structure with *n* outputs to a general 6-dimensional graph, where *n* becomes just a parameter of the Hamiltonian. The new basis is created as follows: the first four states are the ones localized on the vertices depicted, respectively, in green, cyan, purple, and red in Figure 1a. The fifth state is an equal superposition of the internal vertices not yet assigned (the blue ones). Considering a general router with *n* outputs, the sixth state is an equal superposition of the n−1 external vertices painted in yellow. Therefore, the dimensionality reduction method distinguish between the desired and undesired outputs, grouping together identically evolving vertices. Namely, the new orthonormal basis is: (3)|1〉=|green〉|4〉=|red〉|2〉=|cyan〉|5〉=1n−1∑|blue〉|3〉=|purple〉|6〉=1n−1∑|yellow〉.The reduced chiral Hamiltonian is found evaluating the matrix elements of the router Hamiltonian between states of the reduced basis, i.e., Hredkj=kHj, with j,k=1…6 and is given by(4)Hred=01000010βe−iϕ0n−100βeiϕ01n−100010000n−1n−10n−21000010,
where β∈R and ϕ∈[0,2π] are, respectively, the weight and the phase assigned to the link connecting the internal vertices of the input and the target sites. The Hamiltonian in Equation (Equation 4) is the mathematical representation of the effective graph depicted in Figure 1b. The Hamiltonian in Equation (Equation 4) has a fixed dimension independently of the number of outputs *n*, which enters solely as a parameter, paving the way to an easier numerical analysis of the router’s performance for every *n*.

## 3. Routing Classical Information

In this section, we address routing a localized state, whether it represents classical information or the transfer of a spin state in a single-excitation subspace. For simplicity, we will henceforth refer to this scenario as the “routing of classical information”. We analyze the evolution of transition probabilities between two localized nodes, one acting as input and the other as target. The input state is one of the external vertices of the graph in Figure 1a, namely the one depicted in green that corresponds to state 1 of the reduced basis. Once the target state is selected, one needs to apply a phase and/or a weight on the corresponding internal edge. The target state is colored in red in Figure 1a and corresponds to state 4 of the reduced basis. As a figure of merit to evaluate the performances of the quantum router, we use the quantum fidelity [54,55,56,57]. Since we are dealing with pure states, the fidelity between the evolved state and the target one reduces to the transition probability(5)P1,4(t)=|〈4|e−iHredt|1〉|2.On the other hand, the routing probability to the wrong outputs is given by P1,6(t)=|〈6|e−iHredt|1〉|2. Since state |6〉, in the reduced basis, groups together the (n−1) unwanted outputs, then the probability of routing to each of the wrong outputs is P1,6(t)/(n−1). In general, the transition probability will depend on time *t*, the chiral phase ϕ, the weight β, and the number of outputs *n*, showing peaks for some specific configurations. Regarding the dependence on *n*, we have numerical evidences suggesting that it gets weaker as *n* increases (n→∞), converging to some high-*n* limit routing probability. The highest peaks of a (ϕ,β,n) configuration are often very sharp with respect to *t* and ϕ, and will be analyzed in the next section. In this Section, we instead look for broad peaks ensuring a considerable transition probability above the reference value of 0.8 and the robustness of the protocol. We focus on two distinct strategies in which we tune either the chiral phase ϕ or the edge weight β.

### 3.1. Chiral Optimization

In this scenario, we fix the weight at β=1 and maximize P1,4(t) with respect to the chiral phase ϕ∈[0,2π). For a small number of outputs, the protocol is not very effective, because P1,4(t) rarely goes up to 0.8 and oscillates wildly. For n≥10, instead, we see a broad peak emerging slightly above 0.8 at t∼17 and for ϕ∼π, see Figure 2a. This peak may not be the highest, but it is notably broad in both time and phase, making the routing scheme in this regime robust against fluctuations whether they occur in the time of the measurement or in the phase setting. Another important property regarding this peak is that, although a success routing probability of 0.8 is not the maximum, it occurs in a region where P1,6 is very low (∼0.015). This significantly reduces the probability of routing towards undesired outputs and it ensures that when measuring the possible outputs, the walker is most likely found in the target state. This allows ones for the potential repetition of the procedure if necessary. In addition, for even higher values of *n* (more or less starting from 40), a very robust (with respect to ϕ) peak at t∼4 also goes above 0.8.

### 3.2. Edge Weighting Optimization

We now consider the possibility of routing the localized state by optimizing the weight β of the desired link and setting ϕ=0. We can see in Figure 2b the emergence of a very large peak for β≃0.69t. This peak slowly oscillates between 0.95 and 0.99 depending on *n*. In this region, the probability to route into the wrong output P1,6 is low (P1,6≲0.05).

**Figure 2 entropy-27-00498-f002:**
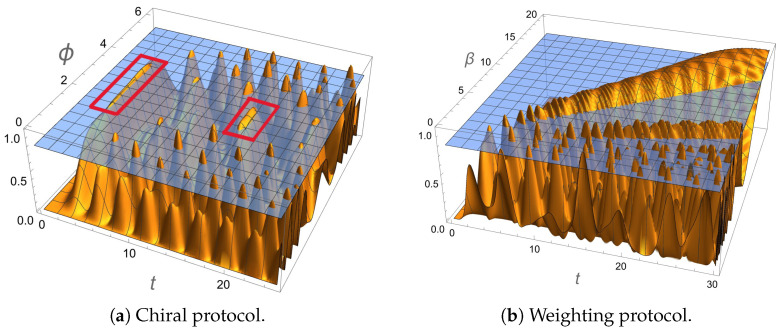
(**a**) The probability of routing a localized input to the target output P1,4(t,ϕ) (in orange) for a number of router outputs n=40 and β=1. The reference fidelity value of 0.8 is represented by the blue surface. The two peaks related to high and robust fidelity (even if not necessarily the highest) are marked in red. (**b**) Routing probability P1,4(t,β) (in orange) for n=50 and ϕ=0. The reference value of 0.8 is represented by the blue surface. In the large weight regime (β>>1), there is large peak emerging above 0.8. Higher times are required if we increase β, but the height of the peak does not vary significantly.

## 4. Routing Quantum Information

In this Section, we address the problem of routing quantum information, i.e., sending an initial superposition state across the network to an output set of nodes, without loosing coherence. In particular, working in the reduced basis, the goal is to send an input state |ψ0〉=α|1〉+1−α2eiχ|2〉 through the router to obtain the target state |w〉=α|4〉+1−α2eiχ|3〉 after a given time. The fidelity between the evolved and target states is still given by the square modulus of their overlap, and depends on time *t* and the parameters of the Hamiltonian: the number of outputs *n*, the chiral phase ϕ, and the weight β. Ideally, we should fine-tune these parameters to ensure a transition probability as close to one as possible, regardless to the input state. When reaching unit fidelity for any state is not possible, two strategies may be employed for optimization. The first is the maximization of the average fidelity with respect to the superposition parameters α,χ, while the second relies on considering the worst-case scenario, i.e., the minimum fidelity with respect to α,χ, and maximizing it.

We analyze the effects of the chiral phase on the routing performances, hence fixing β=1. Since we lack an analytical form for the evolution operator U=e−iHredt, we search for the best configurations through a numerical analysis, by making a choice for the number of outputs *n* and optimizing over *t* and ϕ. Since we aim at scalability with respect to the number of outputs, optimizing for a specific choice of *n* may be regarded as inconsistent with this assumption. However, a configuration that is optimized for a specific number of outputs remains completely valid even if only a fraction of these outputs are utilized. Therefore, when we state that we have a router designed for n=20 outputs, we actually mean that it is capable of handling up to 20 outputs. For this reason, in order to satisfy the request of any number of outputs, we analyze three configurations of the router, with n=20, n=70, and n=106. Regarding the n=20 and n=70 cases, in Table 1, we present some settings that we have numerically found, showing high routing fidelity (≥0.98). We considered both the average fidelity and the minimum one with respect to the initial state parameters α,χ, that actually seem to be locally maximized for slightly different values of the routing parameters.

In the limit of a large number of outputs, the dependence on *n* becomes weaker. In particular, the variation of the fidelity is below ΔnP1,4≈1% when we exceed n=1000. So, we chose the n=106 case as a representative for this high-*n* class of routers. We found a peak of P1,4=0.995 in the worst-case fidelity with respect to the superposition parameters α,χ, for t≈40.068 and ϕ≈4.716, as shown in Table 1. Two remarks are in order, however: Firstly, the routing time is over twice as long as in the other cases, and secondly, it may be challenging to implement a large structure. Eventually, we found three configuration reaching a fidelity exceeding P1,4≈0.98. In Figure 3, we show the behavior of the fidelity as a function of the superposition parameters α and χ, for different router configurations.

## 5. Robustness Against Noise

In the previous sections, we assumed full control on the router parameters (ϕ,n) and the evolution time *t*, which we fine-tuned to give three routing configurations ensuring a transition fidelity P1,4⪆0.98. In this Section, we assess the performance of the quantum router in the presence of fluctuations. Time fluctuations simply correspond to a coarse-graining of the probabilities analyzed in the previous Sections, and we thus focus on the performance of the quantum router when the chiral phase is subject to uncertainty. Specifically, we analyze two models of decoherence: one in which the phase is subject to static disorder and another where the phase is affected by dynamical noise. We refer to these as static and dynamic noise, respectively.

### 5.1. Chiral Phase Subject to Static Noise

We consider a scenario where the chiral phase governing the routing process is fluctuating and should be written as ϕs=ϕ+ϵ, where ϕ represents the optimal phase value and ϵ accounts for the deviation from its ideal value. Here, the phase error ϵ is modeled as a random continuous variable within the range −π≤ϵ<π. Furthermore, we require that the probability of selecting a particular value of ϵ decreases monotonically and symmetrically as it deviates from zero. To satisfy these conditions, we adopt a von Mises distribution [47,58], which naturally captures these characteristics while serving as a circular analogue of the normal distribution. The probability density is:(6)pk(ϵ)=ek cos ϵ2πI0(k).The term I0(k) is the modified Bessel function of the first kind, dependent on *k*, which is the concentration parameter and influences the distribution’s width. Specifically, pk(ϵ)→k→012π and pk(ϵ)→k→∞δ(ϵ). In the latter regime, the von Mises distribution can be well approximated by a normal distribution having variance σ2=1k.

Starting from an initial state ψ0, the evolved mixed state is(7)σout(n,t,ϕ,k)=∫−ππdϵpk(ϵ)U(n,t,ϕ+ϵ)|ψ0〉〈ψ0|U†(n,t,ϕ+ϵ).
where the single realization evolution operator is U(n,t,ϕ+ϵ)=e−iHred(n,ϕs)t. The quantum fidelity F(ρw,σout)≡ρwσoutρw2 between the target state ρw=|w〉〈w| and the final state σout simplifies to(8)F(n,t,ϕ,k)=∫−ππdϵpk(ϵ)|〈w|U(n,t,ϕ+ϵ)|ψ0〉|2.

The effect of static noise is to degrade the previously observed peaks. The degree of reduction is determined by the precision of the phase adopted, i.e., by the concentration parameter *k*. Nonetheless, as a general trend, the first and most robust peak in the fidelity remains present almost independently of the noise applied. This behavior is also independent of the number of outputs *n*. See the upper row of Figure 4 for the behavior of the quantum fidelity for an initial superposition state and concentration parameter values (k=252,258,2). These values were chosen in order to make a fair comparison with the dynamical noise model, as described in the following.

### 5.2. Chiral Phase Subject to Dynamical Noise

To model continuous fluctuations affecting the value of the chiral phase, we consider an Ornstein–Uhlenbeck (OU) noise model [48,49,59]. OU noise is a stochastic Gaussian process {Xt}t governed by the following stochastic differential equation, driven by Wiener noise:(9)dXt=θ(μ−Xt)dt+ΣdWt
where μ is the long-time mean, θ is the mean reversion speed, Σ is the volatility, and dWt is a Wiener increment. A key property of the OU process is that, at any given time, the stochastic variable Xt follows a normal distribution with variance σ2=Σ22θ, provided that the initial condition is not fixed. This feature makes the OU process particularly suitable for modeling noise with temporal correlations, as it naturally describes a system subject to random fluctuations while tending to revert toward a mean value. When OU noise affects the chiral phase, the dynamics of an initial state ρ0=|ψ0〉〈ψ0| over the router is described as an ensemble average over all realizations of the stochastic phase:(10)ρ¯(t)=〈U(n,t,ϕ)ρ0U†(n,t,ϕ)〉ϕ
where U(n,t,ϕ)=Te−i∫0tdt′Hred(n,ϕ(t′)) is the evolution operator generated by the time-dependent Hamiltonian Hred(n,ϕ(t)), T is the time ordering and ϕ(t) is a stochastic phase describing a OU process according to Equation (Equation 9). We again use the quantum fidelity to assess the performances of the quantum router, Fρ¯(t),ρw. Its behavior is displayed in Figure 4 (bottom row) for three different router configurations. To be able to make a comparison with the static noise scenario, we computed the quantum fidelity of the transmitted state using the same initial quantum state |ψ0〉=0.7|1〉−i1−0.72|2〉 and targeting the same final state |w〉=0.7|4〉−i1−0.72|3〉. We analyzed the impact of different noise parameters, considering a unit mean reversion speed θ=1 and testing the volatilities Σ=0.4,0.8,1. For small values of volatility, the fidelity stemming from the OU case follows the unitary evolution, preserving the characteristic peaks over time. However, as the variance of the stochastic phase increases, the fidelity progressively deviates from the unperturbed behavior, leading to a loss of coherence and the degradation of the routing performance.

**Figure 4 entropy-27-00498-f004:**
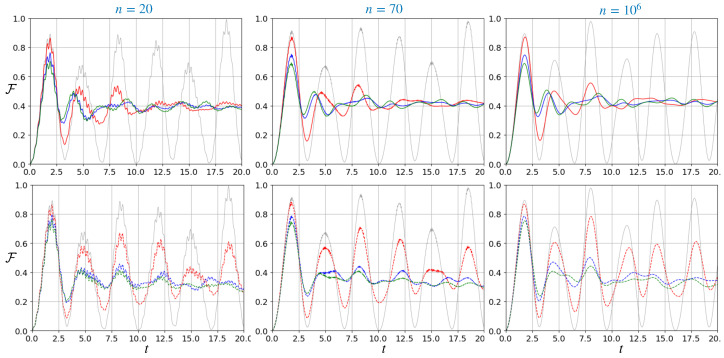
Quantum fidelity between an initial state in the form 0.7|1〉−i1−0.72|2〉 and the target state 0.7|4〉−i1−0.72|3〉 as a function of time, for a different number of outputs *n*, when the chiral phase is affected by static errors (upper row) and OU noise (bottom row). The noise parameters are for the von Mises: k=252 (red), k=258 (blue) and k=2 (green); and the OU: θ=1, Σ=0.4 (red), Σ=0.8 (blue) and Σ=1 (green). The gray thin line is the unperturbed unitary case.

Since the chiral phase can be described by a normal distribution at any given time in both the static and dynamic noise models, we can establish a direct comparison between the two approaches. In our analysis, the distributions were approximated as normal distributions with variances (σ2=225,825,12), ensuring a fair and consistent comparison of the effects of noise in both models (see Table 2).

From Figure 4, we observe that the fidelity appears to be slightly more susceptible to static noise. In both scenarios, however, the quantum fidelity becomes smoother as the number of output ports is increased. Moreover, a key feature that is shared by both noise models is the inevitable degradation of fidelity over time. In the analyzed case, the first peak remains relatively robust, reaching a maximum value of approximately 0.8. In contrast, the subsequent peaks, observed in the noiseless scenario at t∼18.5 and t∼40, become increasingly unreliable under the influence of noise.

## 6. Summary and Conclusions

In this work, we put forward a scalable architecture for high-fidelity quantum routing, enabling the selective transfer of quantum information from one input to one of *n* possible output states. The routing mechanism has been modeled using a chiral continuous-time quantum walk over a structured graph consisting of a complete (n+1)-vertex core, with each internal node connected to an external input or output vertex. Classical information is encoded as a localized excitation, while quantum information is stored in superposition states.

Concerning the routing of localized states, the process is effectively governed by transition probabilities between spatially separated sites, achieved through the application of either a chiral phase or an edge weight to a single link in the graph. This approach can be extended to quantum superpositions, as the routing protocol—regardless of the initial or final state—ultimately relies on introducing a chiral phase on a specific edge of the Hamiltonian. Remarkably, this single operation is sufficient to dynamically select both the sender and the receiver. Assuming precise control over the chiral phase, we demonstrated near-unitary fidelity even in the presence of a large number of outputs, achieving robust performance for cases with several outputs. These findings highlight the scalability of our model, with key properties—such as routing time and fidelity—remaining largely independent of the number of output nodes. This suggests that our protocol provides a promising candidate for scalable quantum information transfer.

Finally, we investigated the impact of noise in the chiral phase, considering both static and dynamical noise models. In both instances, fidelity diminishes over time, rendering the optimization strategies designed for the noiseless scenario ineffective. However, in the cases analyzed, we witnessed the persistence of the first fidelity peak, which remains largely unaffected by the number of outputs or the initial state, despite being lower than the one proposed in the noiseless scenario.

## Figures and Tables

**Figure 1 entropy-27-00498-f001:**
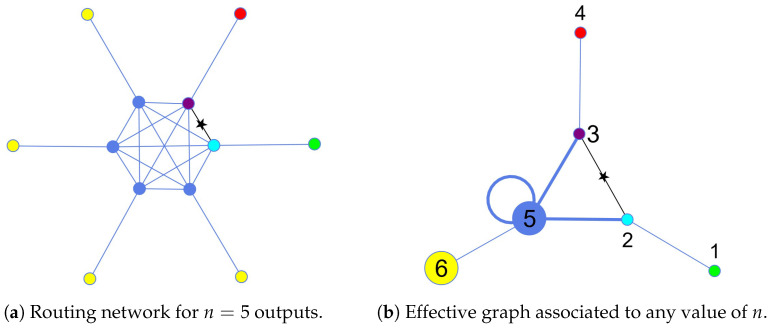
(**a**) Schematic representation of the router, with n=5 outputs. The input state is the green one (classical information/single excitation) or a combination of the green and the cyan ones (quantum information), depending on whether one wants to route localized or superposition states. It follows that the target state can be either the red site or a superposition of the red and the violet. A chiral phase and/or an edge weight (represented by a star) are assigned to the single link connecting the internal nodes corresponding to the input and output vertices (blue and purple). (**b**) An effective graph associated to the router network, representing the reduced basis for the time evolution of the system. We can see the appearance of a self-link over node 5. In addition, all the thick links represent edge weights depending on the value of *n*.

**Figure 3 entropy-27-00498-f003:**
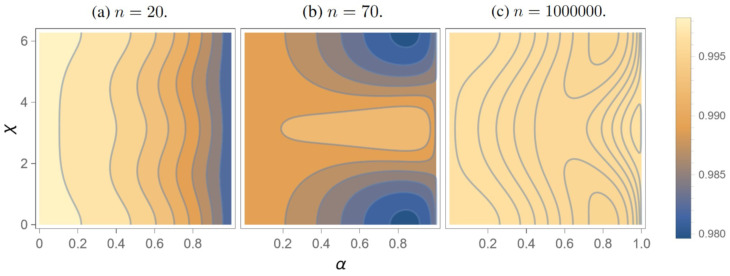
Fidelity for the routing of the superposition state with respect to the parameters α,χ, for (**a**) ϕ=4.712, t=18.550, n=20; (**b**) ϕ=4.758, t=18.397, n=70; (**c**) ϕ=4.716, t=40.068, n=106.

**Table 1 entropy-27-00498-t001:** Configurations presenting high fidelity for the routing of superposition states. We considered both the average fidelity and the minimum fidelity with respect to the superposition parameters α,χ.

*n*	*t*	ϕ (rad)	F(α,χ)
20	18.550	4.712	Favg=0.993
18.523	4.708	Fmin=0.984
70	18.397	4.758	Favg=0.987
18.484	4.765	Fmin=0.976
106	40.068	4.716	Fmin=0.995

**Table 2 entropy-27-00498-t002:** Noise parameters and the variance of the associated Gaussian distribution. In particular, the von Mises distribution with the concentration parameter *k* is approximated by a normal distribution with σ2=1k, while an OU variable can be described by a normal distribution with σ2=Σ22θ. Note that the parameters of the two noise models were chosen in order to be associated to the same Gaussians, allowing a fair comparison between them.

Von Mises (k)	Gaussian (σ)	OU (θ,Σ)
k=252	σ2=225	θ=1,Σ=0.4
k=258	σ2=825	θ=1,Σ=0.8
k=2	σ2=12	θ=1,Σ=1

## Data Availability

Data can be accessed upon reasonable request.

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
