# Peer review of "Scalable Structure for Chiral Quantum Routing"

_entropy, 2025, doi:10.3390/e27050498_

Round 1
Reviewer 1 Report
Comments and Suggestions for Authors
The work done address the issue of state transfer in a graph structure. The authors have demonstrated that a single parameter (chiral phase) can be used to control the state transfer and optimize the fidelity. While the domain and the approach taken is not new, there are certain aspects such as dynamical control using single (or lesser number of) parameter can be regarded as an aspect that controls the scalability and hence applicability of the model. Overall the work is done with scientific and mathematical rigor combined with adequate explanations. However, certain questions pertaining to this work can help increasing its relevance:
- In figure 2, when the authors point out the maximal probability region, what is the relative amplitude of the other peaks. Meaning how are the authors comparing the maximum probability with respect to nearby peaks. If there is not a significant difference, then is it justified to call this Fidelity optimization as such?
- The authors have pointed out the fidelity for the numerical limit of "n" tending to infinity. What happens when the number of input points also increase? Secondly, the authors are treating a cluster as an output node (if I am not mistaken from the eq.3). So when fidelity is taken, is it assumed that a particular cluster is an output node?. Also when local information is transmitted (classical) the authors point out a large n limit. Is there a mathematical form of this limit?
- In case of the noise models, the authors state that the fidelity drops with increasing time apart from the first observed peak. If that is the case, would it be correct to assume that the model is not robust against noise? In that case, if I suppose there is an existing noise in the system, what could be the way to offset the noise using external control parameters if possible?
Overall the work is informative and sheds light in the direction of adoption of practical quantum algorithms for optimizations not just limited to routing. However in order to be scalable, the robustness and decoherence resistance of the model could be sharpened with respect to time. I would advocate for this work to be given a place in the journal owing to answers to the above issues.
Reviewer 2 Report
Comments and Suggestions for Authors
This paper presents a scalable architecture for routing quantum and classical information in a quantum network using continuous-time chiral quantum walks on a structured graph. It demonstrates that tuning a single phase or edge weight enables information routing with near-unit fidelity and analyzes the system's robustness under both static and dynamic noise. The work extends Ref. [37] by allowing for a large number of nodes in the network. I believe the paper could be slightly improved if the authors address the following points:
1. The evolution operator does not include an explicit parameter γ; it appears to be implicitly absorbed into the time variable due to unit conventions. If that is the case, then γ is effectively fixed and independent of n. Why doesn’t the paper explore the possibility of using a γ that depends on n?
2. The statement in Section 4, “Since we lack an analytical form for the evolution operator,” is misleading. Does it mean “we could not find” or “it is not possible to find”? The authors probably meant something else. The reduced Hamiltonian is a 6×6 matrix with analytical dependence on n, so the evolution operator is analytical. In principle, the exponentiation of the Hamiltonian can be computed analytically via spectral decomposition. Even if this is nontrivial, analytical results are likely possible, especially in the n-to-infty limit.
3. References 2 and 6 seem to be repeated.
4. 'orhonormal' missing a t must be orthonormal in line 132
Round 2
Reviewer 1 Report
Comments and Suggestions for Authors
The authors have done their due diligence in answering the concerns. While the present work leaves some gap, I believe that those gaps have been clarified as to being within or outside the scope of the present work. Further these gaps have also been addressed as what could possibly be in the scope of future directions extending the work to more system agnostic design and implementation. In lieu of this, I would recommend the work to be accepted in the journal in the present form and with all queries answered.